# Biodegradable Poly(butylene succinate) Laminate with Nanocellulose Interphase Layer for High-Barrier Packaging Film Application

**DOI:** 10.3390/foods12224136

**Published:** 2023-11-15

**Authors:** Martins Nabels-Sneiders, Anda Barkane, Oskars Platnieks, Liga Orlova, Sergejs Gaidukovs

**Affiliations:** 1Institute of Polymer Materials, Faculty of Materials Science and Applied Chemistry, Riga Technical University, P. Valdena 3/7, LV-1048 Riga, Latvia; martins.nabels-sneiders@rtu.lv (M.N.-S.); anda.barkane@rtu.lv (A.B.); oskars.platnieks_1@rtu.lv (O.P.); 2Institute of Materials and Surface Engineering, Faculty of Materials Science and Applied Chemistry, Riga Technical University, P. Valdena 3, LV-1048 Riga, Latvia; liga.orlova@rtu.lv

**Keywords:** composite, nanofibrillated cellulose, morphology, ultraviolet–visible spectroscopy, sustainable, green material, moisture vapor transmission rate

## Abstract

In response to rising concerns over the environmental and human health ramifications of polymers derived from petroleum, particularly in the food packaging industry, research has pivoted towards more sustainable materials. Poly(butylene succinate) (PBS), selected as the polymer matrix, stands out as one of the most promising bio-based and biodegradable polymers suitable for film blowing and lamination. A layered spray-coating technique was employed to apply 1, 5, 10, and 20 layers of nanofibrillated cellulose (NFC) between blown PBS films, creating a three-layer laminate structure. NFC sourced from minimally processed hemp stalk waste highlights the potential for minimizing environmental impact. The water vapor transmission rate (WVTR) of these films, a critical parameter for food packaging, was assessed in a controlled environment at 38 °C and 90% relative humidity over a period of two months. The integration of a single NFC layer, constituting 0.35% of the composite’s weight, was observed to significantly reduce the WVTR by up to 5.5-fold. It was noted that higher NFC layer counts above 10 reduced the adhesion within the laminate layers. Morphological assessments showed that the number of structural defects increased with a higher count of NFC layers. As the count of NFC layers increased, the optical transparency of the laminates dropped from approximately 65% to 25% in the visible light spectrum. Notably, by weight percent, NFC proved to be an effective barrier even without chemical modification. The developed laminates stand out as a viable, green option for food packaging, offering a sustainable and renewable solution.

## 1. Introduction

Food packaging, predominantly composed of single-use plastics, significantly contributes to global waste [1], adding to the volume of municipal solid waste and leading to detrimental environmental and human health effects [2,3]. This packaging, often discarded after one use, is not only a source of non-biodegradable waste but also a driver of microplastic contamination, which is closely linked to our reliance on conventional disposable plastics [4,5]. The challenges of disposing and recycling such materials are substantial, posing a severe obstacle within the framework of sustainable development [6,7]. Recycling single-use plastic packaging is not only laborious and energy-intensive but also remarkably inefficient [8]. Therefore, there is an urgent need to transition to alternatives that meet our packaging needs while safeguarding societal and environmental welfare. This shift requires a fundamental change in approach, favoring sustainable and biodegradable materials, and is further underscored by the vast resource consumption and emissions associated with the life cycle of packaging materials [4,8], as well as the economic losses from their disposal. Innovations in sustainable packaging and stronger regulatory measures are crucial to addressing these multifaceted issues.

A possible replacement for plastics derived from petroleum is bio-based and biodegradable polymers [9,10]. These include polylactic acid (PLA), polyhydroxyalkanoates (PHA), poly(butylene succinate) (PBS), and their copolymers or blends. When it comes to packaging, exceptionally high standards must be met to avoid contamination and preserve product quality [11]. PBS has good mechanical properties, biodegradability, and excellent thermal processability and is non-toxic, enabling its use for food storage [12,13]. Vytejčková et al. showed that PBS films are suitable for raw and smoked poultry meats [14], while Číhal et al. showed that PBS provided good barrier properties for common aroma compounds in foods [15]. PBS’s low melting point of around 115 °C presents significant advantages when it comes to vulnerability to thermal degradation during thermal processing. Unfortunately, PBS has drawbacks, most notably concerning its moderate water vapor and poor oxygen barrier properties [16]. The incorporation of fillers can improve these barrier properties [17,18]. A particularly popular choice is the use of nanoclay for the enhancement of barrier properties [19]. Different nanofillers have been explored to enhance the barrier properties, including Janus nanosheets [20], chitin whiskers [21], nanocrystalline cellulose (NCC) [22], and nanofibrillated cellulose (NFC) [23].

Nanocellulose is advantageous for food packaging due to its sustainable profile and compatibility with foodstuffs [24]. NFC stands out as a superior choice over NCC for food packaging material preparation. NFC creates a dense, entangled fibrous network that provides exceptional barrier qualities against gases, enhancing product longevity and safety [25]. Moreover, the production of NFC is more cost-efficient and environmentally friendly, involving less energy and fewer chemicals than the production of NCC [26,27,28]. The literature suggests that antimicrobial agents form good synergy with nanocellulose [25,29]. Melt blending of nanocellulose and composites presents drawbacks like poor dispersion and agglomeration, which require increasing filler amounts to relatively high loadings [21,22,30]. Similarly, casting films from solvents is not suitable for the manufacturing of industrial packaging films.

A coating layer or lamination can serve as a simple, effective, and cheap solution to improve barrier properties. Yook et al. reported that using NFC coatings can significantly enhance the barrier properties of papers [31]. The authors noted that fibril size is directly related to achieved properties and that the hydrophilic properties of NFC should be considered and mitigated. Bideau et al. proposed the use of (2,2,6,6-Tetramethylpiperidin-1-yl)oxyl (TEMPO) oxidized cellulose nanofibers and polypyrrole (PPy) as coatings on paperboard [32]. The authors reported that coatings under 100 μm reduced air permeability by more than 10-fold, making them potential replacements for conventional lamination with non-degradable plastics. Our previous work demonstrated that cast hemp papers can be laminated with biodegradable polymers like PBS to produce durable composites which can work as replacements for conventional laminated paperboard [33]. A novel approach which improves the properties of PBS can be examined by combining the ideas of lamination and thin layers of coated nanocellulose as a barrier layer.

This study aims to prepare laminated PBS films with a thin layer of nanocellulose coating between the PBS layers. With a variation of nanocellulose coating layers from 1 to 20, the morphology, layer adhesion, thermal stability, UV-Vis transmittance, and water vapor barrier properties were explored. The multilayer laminate appears to be an excellent candidate for preparing high-barrier biodegradable packaging. The PBS/NFC laminates are designed for short-term food storage applications, which include, but are not limited to, squeezable foods, dry goods, bread, meat, and vegetables. The present study insufficiently explores critical facets of food safety in relation to these laminates. It is essential to conduct in-depth research examining aspects like the impact of refrigeration temperatures, microbial kinetics, and oxygen permeability characteristics, among others.

## 2. Materials and Methods

### 2.1. Materials

BioPBS™ poly(butylene succinate) (PBS) grade FZ71PB^®^ is a thermoplastic polymer that is 50% bio-based (from bio-based succinic acid) and compostable, and its suggested applications include film blowing, injection molding, and lamination. It has a melting point of 115 °C, a melt flow index of 22 g/10 min (2.16 kg at 190 °C), and a density of 1.26 g/cm^3^. Hemp stalks of the Santhica 27 variety were used as a cellulose source. Sodium hydroxide (NaOH), potassium sulfate (K_2_SO_4_), and calcium chloride (CaCl_2_) were purchased from Merk KGaA (Darmstadt, Germany).

### 2.2. Preparation of Nanocellulose

Hemp stalks were prepared using two grinding cycles in a Retsch SM300 cutting mill with 4.00 mm and 0.25 mm sieves. The mill’s rotating speed was set at 1500 rpm, and it was manually fed throughout the procedure. An alkaline treatment was used to increase the cellulose content. Hemp was heated at 80 °C for 3 h while being continuously stirred in a 5 wt% NaOH solution (1:8 cellulose to NaOH solution). Then, the solution was replaced and continuously stirred at room temperature for 12 h. Following processing, hemp particles were filtered and washed with distilled water until their pH levels were neutral. The NaOH treatment reduces the biomass weight by about 20 to 30%. The hemp suspension concentration was adjusted to 1.0 wt%. The microfluidization method was employed to prepare NFC [34]. The resulting aqueous suspension was passed through a microfluidizer (Microfluidic, LM20, Westwood, MA, USA) with chamber H210Z (200 m). Each suspension underwent five passes to achieve the optimal defibrillation level. The pressure was set at 10,000 psi for the initial pass, which was increased to 30,000 psi for subsequent passes. In our previous study [35], where an in-depth analysis of treated hemp stems was carried out, the average diameter of the nanofibrils was 70 nm, and the cellulose content was 89%. Figure 1 shows that the prepared NFC includes some microfibrils which contribute to an increased average diameter; hence, it may be described as micro/nanofibrillated cellulose (MNFC).

### 2.3. Blown Film Extrusion 

A Labtech extruder (model LTEM20–48, Samutprakarn, Thailand) equipped with a coextrusion-blown film line (model LF400) was used to prepare the PBS films. Barrel temperatures were optimized between 115 and 130 °C. The extrusion head was set to 130 °C, and the line speed was adjusted between 8 and 10 rpm. Air cooling was used at an ambient temperature of 20 °C. The thickness of the produced blown film is about 100 ± 10 µm. 

### 2.4. Sample Preparation—Coating and Lamination

The spray-coating method for applying cellulose suspension to PBS film was used. A spray coater brush and air compressor were set up with 1.5 bar pressure, 135° spraying angle, and 15 cm spraying distance. The concentration of NFC/water suspension was adjusted to 0.8 wt% after process optimization. One to twenty layers were applied to square-sized samples (10 × 10 cm). The drying time between layer applications was set to 5 min. After the NFC-coated films were prepared, the NFC layer was sealed between the PBS films by adding one more uncoated PBS film on top using compression molding (112 °C, 2 min). Laminated films were rapidly cooled between stainless steel plates. Four samples and one reference sample were prepared for measurements using the procedure. The sample abbreviations and applied NFC coating layers are summarized in Table 1. The NFC weight content within the laminate was determined using an analytical balance based on measurements from five parallel samples. Each spray-coated layer deposited between 0.706 and 0.917 g/m^2^, with a higher layer count resulting in less mass deposited. 

### 2.5. Characterization Methods

Optical microscopy (OM) images of coatings were obtained with Leica DMR (Leica Microsystems) (Wetzlar, Germany) at 100× magnification. The image processing was performed with Leica Image Suite™ and Fiji software ImageJ 1.53J. Atomic force microscopy (AFM) topography images of the coatings were obtained with an atomic force microscope (Smena, NT-MDT, Moscow, Russia) equipped with a HA_NC (ETALON) tip in semi-contact mode. Laminate cross-cut surface images were obtained using scanning electron microscopy (SEM). The FEI Nova NanoSEM 650 Schottky field emission scanning electron microscope (FESEM) with a 10 kV acceleration voltage was utilized, and the samples were mounted on electrically conductive double-sided carbon tape. Diluted hemp NFC suspension was applied to the copper grid (mesh 200) and left to evaporate at ambient room temperature. The laminate specimens were fractured after cooling in liquid nitrogen. No sputter coating was applied.

Adhesion T-peel tests were performed using the Tinius Olsen model 25ST (Horsham, PA, USA). A 5 kN load cell with a 1 mm/min testing crosshead speed was employed in the experiment. Each laminate had five parallel measurements at 20 °C and 40% relative humidity. Specially prepared strips 10 mm wide and about 80 mm long were used (laminated to approximately 40 mm of their length). 

On TG50 equipment (Mettler Toledo, Horsham, PA, USA), using a heating rate of 10 °C/min, thermogravimetric analyses (TGA) were carried out. In the air environment, samples weighing about 10 mg were heated between 25 and 700 °C. From the weight loss heating curves, the material’s thermal stability was assessed. Using the original Mettler program, the weight loss was computed in accordance with ASTM D3850. The isothermal method (40 °C, 16 h) was also used to study the water evaporation process’s kinetics and cellulose’s impact on water absorption. Before the isothermal test, the samples were submerged in distilled water for 24 h at 20 °C and wiped with a tissue after removal from the water. 

A SolidSpec3700 UV-VIS-NIR Shimadzu (Kyoto, Japan) spectrophotometer was used to measure UV-Vis absorbance from 300 to 740 nm. A white BaSO_4_ plate was used as the reference plate for all measurements. The final spectra were created by combining three parallel measurements.

A laboratory balance test was used to measure the water vapor transmission rate (WVTR). Glass jars with a diameter of 5 cm were filled with anhydrous calcium chloride (CaCl_2_). Using epoxy resin as an adhesive, the jars were sealed with the produced laminates. The jars were inserted in the desiccator with a controlled relative humidity (RH) of 90% (in a potassium–sulfate-saturated salt solution environment). The sample weight increases were measured every 24 h (except for weekends) over a period of two months. The results were averaged from three parallel measurements for each sample. 

## 3. Results and Discussion

### 3.1. Morphology

The morphology of the sprayed NFC layer and prepared laminate composite samples was investigated by optical microscopy (OM), atomic force microscopy (AFM), and scanning electron microscopy (SEM) accordingly (see Figure 2). OM in Figure 2a shows fiber accumulation on the PBS film surface. Sample cPBS 1 clearly shows that the spraying method used for one layer is insufficient to form a homogeneous uninterrupted cellulose fiber coating, as uncoated spots can be seen. The OM images with increasing layer counts (5, 10, and 20) reveal the evolution of the cellulose fiber deposition. Samples cPBS 5, cPBS 10, and cPBS 20 all show a homogeneous and uninterrupted fiber layer. With a higher layer count, the density of the packed network increases, as well as fiber entanglement and surface roughness (indicated by the inability to obtain a 20-layer sample in focus).

Surface structure and roughness were further investigated with AFM. Figure 2b, with the generated 3D topography images, clearly shows how surface depth increased almost three times with the increasing layer count. However, AFM can measure the depth of the cellulose layer if an edge of the coating is studied. For a more accurate assessment of the layer depth, SEM was used. The AFM image of cPBS 1 reveals a surface with a partial NFC coating. The image of the cPBS 5 surface shows a fully formed coating with local depth variations. A pronounced layering pattern of coating can be observed for the cPBS 10 sample. AFM and OM show relatively even NFC distribution and visually optimal performance. The cPBS 20 sample exhibits a rough surface structure, showing formations of large fiber-like structures, while smaller fibrils become less pronounced. It can also be seen that cellulose tends to agglomerate together when applying a high number of layers, forming structural voids.

SEM images (Figure 2c) of laminate samples exclude the cPBS 1 sample, which was hard to produce due to the contrast and burning of cellulose at a higher voltage. Coating layer dimensions are 3.3–5.6 µm, 5.1–9.4 µm, and 31–39 µm for cPBS 5, cPBS 10, and cPBS 20, respectively. There is good scaling between 5 and 10 layers, showing a two-fold increase matching the layer increase. The NFC layer is noticeably thicker for cPBS 20 than the other samples. Both cPBS 5 and cPBS 10 form dense layers without visible defects (unless layer thickness uniformity is considered). But cPBS 20 forms a low-density layer with the presence of voids (as air gaps). This could be related to the drying conditions and the tendency of cellulose to migrate and agglomerate after applying each layer. Thus, as the coating layer gets thicker, the agglomerated structures and voids become larger and trap air inside the structure.

Similar observations with solvent-promoting particle agglomeration were observed when applying spray-coated wax coatings [36]. The higher layer count of spray coating did not scale with surface properties, and too many layers created defects in the structure. Tabak et al. used spray-coated NFC layers to prepare melt-extruded polymer composites [37]. In their study, the authors presented about 60 µm thick NFC coatings, which showed clear layer separation and relatively loose structure. Herrera et al. applied thin nanocellulose coatings using spin- and dip-coating methods [38]. The authors demonstrated that dip-coatings formed 9 and 23 μm thick layers with dense structures obtained from 1 wt% nanocellulose suspension with 5 and 10 layers, respectively.

### 3.2. Adhesion T-Peel Test 

The T-peel test force–displacement curves for laminates are shown in Figure 3. Due to partial coating formation, cPBS 1 did not yield peelable laminate layers. Relatively close values of cPBS 5 and cPBS 10 match the morphology images, which indicate a similar structure of dense NFC interphase between PBS films, while lower force values for cPBS 20 match the observation of a loose NFC coating structure. This could explain the weaker adhesion due to a reduced contact surface. The achieved maximum interfacial peel strength of 0.08 N/mm is a relatively satisfactory result, indicating that further investigations are still needed to enhance laminate adhesion. For comparison, Liang et al. showed that an interfacial peel strength of 0.06 N/mm can be achieved between aluminum foil and polypropylene [39]. The authors showed that T-peel strength can be enhanced 10-fold using an oxygen group containing grafted polypropylene copolymer.

### 3.3. UV-Vis Spectroscopy

Figure 4a presents the UV-Vis transmittance spectra of the laminate samples, and Figure 4b provides a comparative analysis of transparency through optical imaging. Given that PBS is semi-crystalline, its transparency is naturally limited [12], which is particularly evident in thicker films, explaining the initial 60 to 70% transmittance range for visible light. The laminates have an approximate thickness of 200 µm, which is increased by up to 10 µm with the inclusion of NFC layers (the exception is cPBS 20, which showed a significant increase in thickness up to 40 µm). It is important to note that the transmittance values presented have not been adjusted for thickness variations; therefore, any decrease in transmittance is partly attributable to the increased thickness of the laminate. 

The introduction of a single NFC layer, approximately 1 µm in thickness, resulted in a reduction in the visible light transmittance of around 5%. Meanwhile, the transmittance spectra of cPBS 5 showed only marginal differences compared to those of cPBS 1. However, beyond five layers, a clear trend emerged, with a substantial decrease in transmittance; cPBS 20’s transmittance dropped to about 20% in the visible spectrum. While reducing UV-Visible transmittance might detract from the visual appeal, it proves advantageous for food packaging applications. Exposure to UV light can significantly compromise food quality through the generation of free radicals [40]. Furthermore, the incorporation of cellulose notably affected light transmittance across the spectrum, with a pronounced disparity at different wavelengths—transmittance at longer wavelengths (700 nm) was about 20% higher than at shorter wavelengths (400 nm). The absorbance of laminates sharply increased at wavelengths around 370 nm. The sharp rise in absorbance suggests that the material undergoes a particular electronic transition or is in the presence of an absorbing species. Łopusiewicz et al. reported only around 10% transmittance in the visible light range for similar thickness PBS films [40]. This could be attributed to different processing and transcrystallization conditions, as the authors used similar grade PBS.

### 3.4. Thermogravimetric Analysis

The thermogravimetric analysis (TGA) curves are presented in Figure 5a. Laminates incorporating cellulose layers exhibit a thermal degradation profile comparable to the PBS 0 sample. The introduction of cellulose resulted in a slight decrease in the initial degradation temperature, lowering it by up to 15 °C. The thermal degradation only starts at temperatures above 250 °C, well above the processing temperature of laminates. Furthermore, TGA data confirm that the laminate fabrication process successfully eliminates residual water from the structure. The cPBS 20 displayed a distinct final phase in its degradation curve which can be ascribed to char formation, a phenomenon that is frequently reported in the literature [41]. 

The isothermal heating (40 °C) graph for laminates submerged in water for 24 h is presented in Figure 5b. The cPBS 0 showed about 4.5% weight loss, while the cPBS 10 lost about 6.3% of its weight. From these data, it is inferred that each additional NFC layer contributed roughly to a 0.18% increase in water absorption. cPBS 0 reached stable weight after about 12 h and cPBS 10 after 15 h. The weight increases due to NFC layers indicate that water can effectively migrate through PBS films. 

### 3.5. Water Vapor Transmission Rate

Figure 6 depicts the variations in water vapor transmission rate (WVTR) over a period of 57 days. The cPBS 0 laminate, lacking an NFC layer, exhibits the highest WVTR values, ranging from 26 to 30 g/m^2^/day. Over time, these values decline and stabilize around 26 g/m^2^/day, a trend that may be ascribed to water vapor-induced structural changes. Such changes could potentially reduce the crystallinity of PBS and alter the structure and intermolecular bonds within PBS chains [42]. Although changes in PBS are within the bounds of reasonable measurement error, they are noteworthy.

The lowest measured WVTR was registered for cPBS 1, closely followed by cPBS 5. Comparative data after 20 and 50 days indicate that cPBS 1 exhibits WVTR values that are 82% and 49% lower, respectively, compared to cPBS 0. As the results for cPBS 5 are comparable to those of cPBS 1, there is potential for further research into optimizing WVTR with NFC layer count. cPBS 20 shows a 9–23% reduction in WVTR compared to cPBS 0. Both cPBS 10 and cPBS 20 achieve a constant WVTR, whereas the other samples exhibit slight ongoing changes in their value trends after 57 days. These observations imply that the structural changes in PBS might have a more pronounced effect on the properties of films with fewer NFC layers applied.

For laminates with an NFC, an increase in the number of cellulose layers corresponds to higher WVTR values. Since cellulose is insoluble in water and forms robust hydrogen bonds, it extends the diffusion path of water molecules through the laminate. As a hydrophilic polymer, cellulose swells upon moisture absorption, decreasing the density of the layer, generating structural defects, and creating additional diffusion pathways, which accounts for the observed increase in WVTR over time [43]. Morphological observations suggest that a higher number of layers leads to a less dense laminate structure, which also contributes to the observed differences between laminate WVTR. Consequently, cPBS 20 reaches equilibrium more rapidly due to its structure having numerous defects, like voids, which allow for quicker diffusion. This swelling phenomenon and aqueous diffusion through polymers have been previously investigated and modeled [44], which also accounts for deviations from equilibrium.

It was reported that a PLA/PBS 90/10 wt% blend (using PBS from the same manufacturer) achieved WVTR of 37.35 g/m^2^/day (97% RH at 25 °C) for a film with 225 µm thickness (cPBS 0 is about 200 µm) [45]. In a different study, 50 µm thick PBS (using PBS from the same manufacturer) film was reported to have 114.5 g/m^2^/day WVTR (ASTM E96, 97% RH) [46]. Xu et al. reported a WVTR of 83.8 g/m^2^/day (90% RH and 37.8 °C) for PBS films with a thickness of 50 µm, which was further reduced to 49.4 g/m^2^/day by the addition of 3 wt% of NCC [21]. Therefore, our findings are consistent with the expected range and align with values reported in other studies. Discrepancies can be attributed to variations in the grade of PBS, film thickness, testing environment, and film processing techniques.

## 4. Conclusions

NFC was successfully integrated as an interphase between PBS laminate layers. Morphology images revealed that 1 coating layer of NFC formed a partial coating with visible defects in the structure, followed by a dense interphase layer of NFC between 5 and 10 layers, while 20 coating layers resulted in a loose NFC structure with visible defects. The NFC interphase layer directly affected layer adhesion. One coating layer (cPBS 1) did not separate consistently (strongly bonded layers), while cPBS 20 had relatively low adhesion between cellulose and PBS. The thermal stability of films was not significantly effected by the presence of NFC. Analysis of isothermally heated wet laminates revealed that each NFC layer increased the water uptake of the laminate by 0.15% to 0.20% (from the dry sample weight).

UV-Vis spectroscopy revealed that PBS as a semi-crystalline polymer has a transparency below 75%, reduced by up to 10% by applying 5 NFC layers. Water vapor transmission rate studies over the period of two months revealed that the most effective approach is using fewer NFC layers, with cPBS 1 showing from 49% to 82% lower values than cPBS 0 (up to a 5.5-fold improvement). Over the testing period, the cellulose layer’s ability to enhance WVTR saw a gradual decrease, which was attributed to cellulose swelling. One NFC layer contributed to about 0.35 wt% of PBS laminate with a thickness of 200 μm. The remarkably low amount of NFC required to enhance barrier properties clearly demonstrates the efficacy of the lamination approach. 

As the best improvements in WVTR were achieved with the low amount of spray-coated NFC layers, further follow-up research should be conducted to optimize the laminates. In addition, more research is needed to determine various parameters associated with food safety. To improve the concept, antimicrobial additives could be incorporated into the NFC layer. PBS/NFC laminates are primarily designed for short-term food storage, serving as sustainable alternatives to traditional fossil-based plastics. 

## Figures and Tables

**Figure 1 foods-12-04136-f001:**
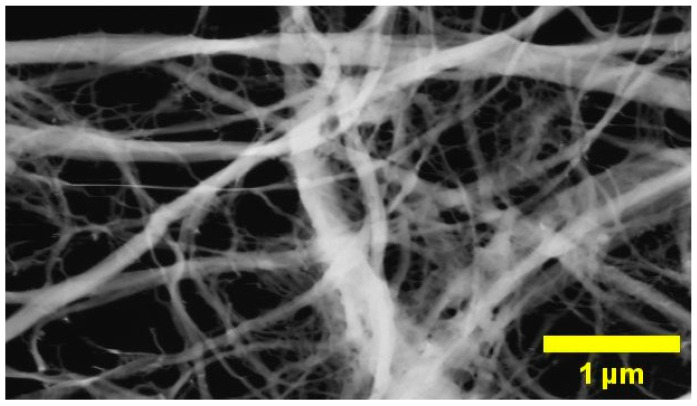
Scanning electron microscopy image of nanofibrillated cellulose.

**Figure 2 foods-12-04136-f002:**
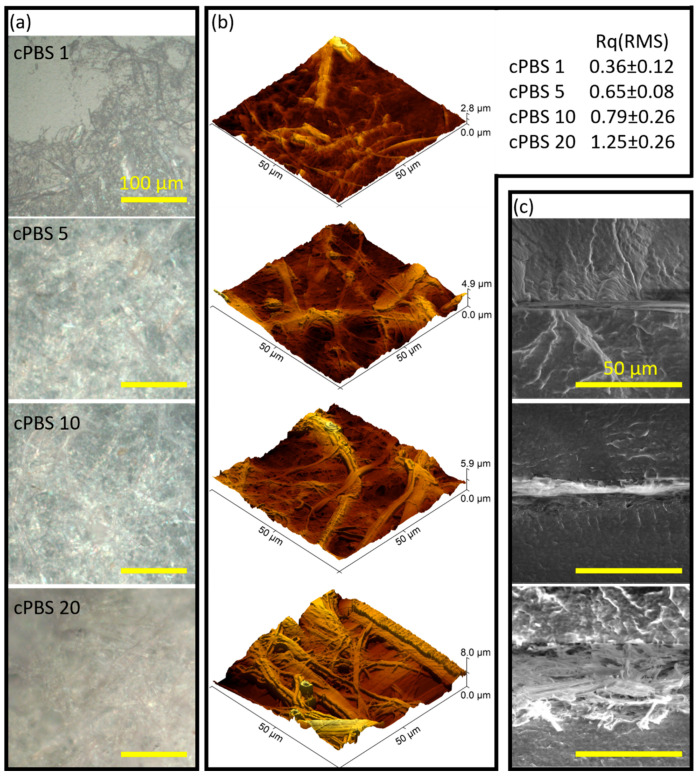
(**a**) Optical microscopy and (**b**) AFM 3D topography images of 1-, 5-, 10-, and 20-layer NFC-coated PBS films, and (**c**) SEM micrographs of 5-, 10-, and 20-layer NFC laminate cross-sections.

**Figure 3 foods-12-04136-f003:**
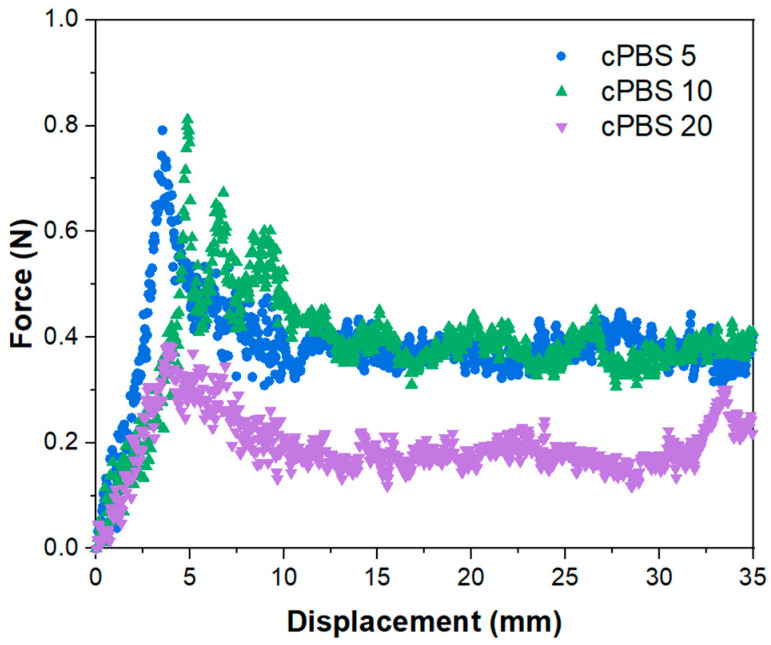
Adhesion T-peel test force–displacement curves for 5-, 10-, and 20-layer laminates.

**Figure 4 foods-12-04136-f004:**
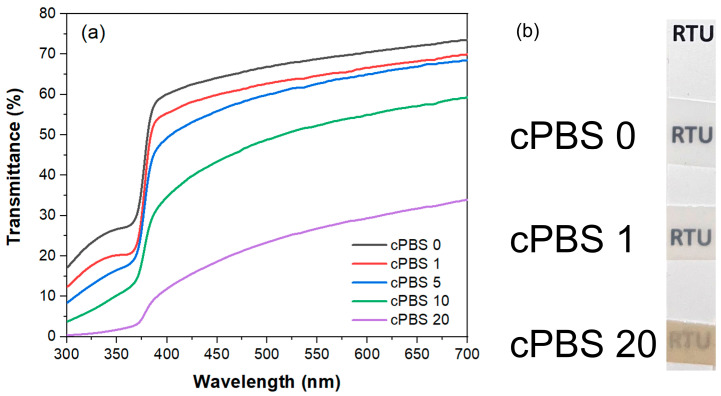
(**a**) UV-Vis transmittance spectra in the wavelength range 300–700 nm for laminates. (**b**) An optical image showcasing the transparency levels of the laminates: cPBS 0, cPBS 1, and cPBS 20.

**Figure 5 foods-12-04136-f005:**
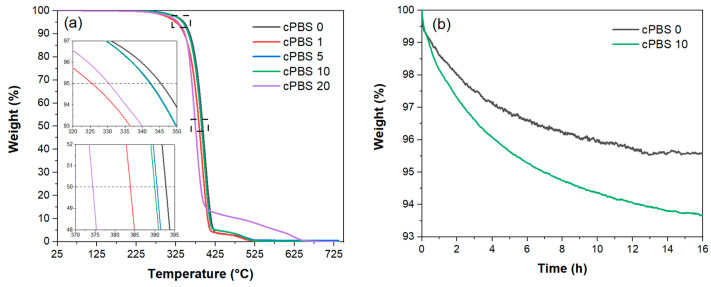
(**a**) TGA thermogram for laminates and (**b**) TGA thermogram in isothermal mode for wet laminates.

**Figure 6 foods-12-04136-f006:**
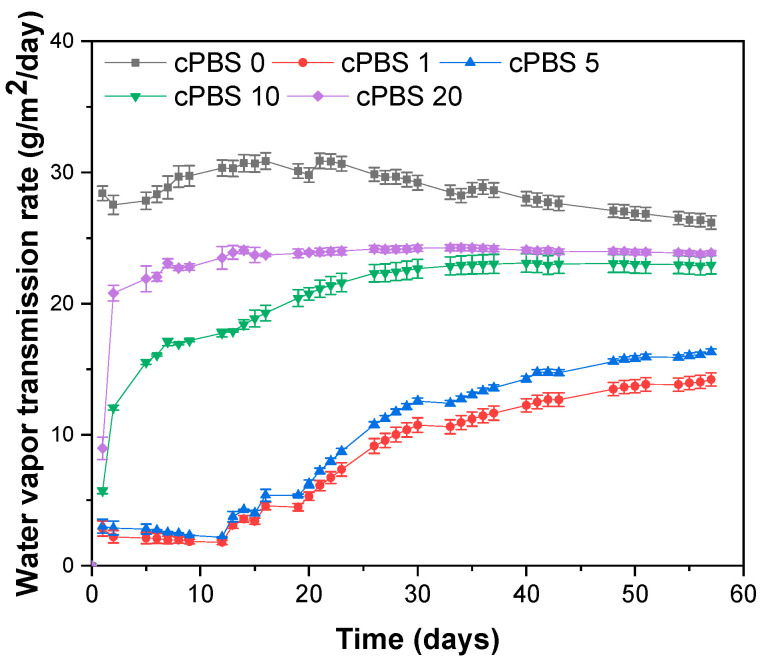
Changes in water vapor transmission rate (WVTR) for laminates over the period of 57 days.

**Table 1 foods-12-04136-t001:** The prepared laminate samples.

Sample Abbreviations	Spray-Coated NFC Layers	NFC Content (wt%)
cPBS 0	-	-
cPBS 1	1	0.346 ± 0.057
cPBS 5	5	1.575 ± 0.264
cPBS 10	10	2.913 ± 0.407
cPBS 20	20	4.702 ± 0.101

## Data Availability

The data presented in this study are available on request from the corresponding author.

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
