# Peer review of "Biodegradable Poly(butylene succinate) Laminate with Nanocellulose Interphase Layer for High-Barrier Packaging Film Application"

_foods, 2023, doi:10.3390/foods12224136_

Round 1

Reviewer 1 Report

Comments and Suggestions for Authors

The manuscript titled "Biodegradable Poly(Butylene Succinate) Laminate with Nanocellulose Interphase Layer for High Barrier Packaging Films Application" needs improvement in the following aspects.

1. The abstract has to be rewritten by including the salient experimental results.

2. The introduction section is weak in terms of portraying the novelty of the identified problem. It can be improved by referring to some of the following recently published articles. https://doi.org/10.1016/j.carbpol.2020.117479, https://doi.org/10.1002/9783527837304.ch1https://doi.org/10.1016/j.foodhyd.2019.105411.

3. There are some older articles listed in the reference sections. Authors may replace them with recently published articles.

4. Sections 2 and 3 have to be swapped. The manuscript is not in proper order. 

5. What is the reason for the gradual loss of weight by cPBS 20 when compared with its counterparts? Kindly justify.

Comments on the Quality of English Language

English language needs a thorough proofreading

Author Response

The manuscript titled "Biodegradable Poly(Butylene Succinate) Laminate with Nanocellulose Interphase Layer for High Barrier Packaging Films Application" needs improvement in the following aspects.

- Thank you for your comments. We have improved our manuscript according to your suggestions.

  1. The abstract has to be rewritten by including the salient experimental results.

-We have rewritten and improved the abstract.

  1. The introduction section is weak in terms of portraying the novelty of the identified problem. It can be improved by referring to some of the following recently published articles. https://doi.org/10.1016/j.carbpol.2020.117479, https://doi.org/10.1002/9783527837304.ch1, https://doi.org/10.1016/j.foodhyd.2019.105411.

- Thank you! We have enhanced the research with relevant articles.

  1. There are some older articles listed in the reference sections. Authors may replace them with recently published articles.

- We have added some new references and tried to use the newest and most relevant ones.

  1. Sections 2 and 3 have to be swapped. The manuscript is not in proper order. 

- Yes, thank you for noticing; we have made these changes.

  1. What is the reason for the gradual loss of weight by cPBS 20 when compared with its counterparts? Kindly justify.

- Slightly improved the clarity and writing of this section and added  “The cPBS 20 displayed a distinct final phase in its degradation curve that can be ascribed to char formation, a phenomenon that is frequently reported in the literature.”

Reviewer 2 Report

Comments and Suggestions for Authors

The experimental article "Biodegradable Poly(Butylene Succinate) Laminate with Nanocellulose Interphase Layer for High Barrier Packaging Films Application" describes the process of obtaining a multilayer coating intended for food products, and includes the results of ongoing research by these authors. The manuscript describes the preparation of a fibrillated cellulose-containing product from hemp stems, called “nanocellulose” by the authors, which is proposed to be applied layer-by-layer to PBS. As conceived by the authors and similar to the research of Xu [16. doi:10.1007/s10965-019-1783-8], the new laminate, while retaining all its valuable properties for food packaging, should acquire a new quality, specifically reduce the water vapor transmission rate and become “High Barrier Packaging Films”. Unfortunately, in the author’s interpretation, instead of commercial nanocrystalline cellulose and the technology of integration into PBS, it is proposed to introduce the resulting fibrillated cellulose-containing product from hemp stems with a main substance content of 89% into the coating using the “layer-by-layer” method. As it turned out, the effect of reducing the water vapor transmission rate can be achieved by applying from one to five layers, but at the same time the packaging loses its transparency and it is necessary to control the peeling of this packaging. One can recognize the honesty of the researchers-authors of the manuscript, but this manuscript in its presented form poorly corresponds to the Foods rating publication and requires serious revision. Below are comments and questions.

Comments and questions that need to be answered with the text of the article:

1. It is recommended to include in the title of the article, in the annotation, and in conclusion, the definition of these films as “food”, specifically, “intended for food packaging.” Example of the title “Poly(butylene succinate) with an interfacial layer of nanocellulose as high-barrier food packaging films.”

2. Recommend to indicate in the annotation the types of food products that can be packaged with the described coating. Proposal "NFC can serve as an effective barrier without the use of chemical modification methods." can be excluded because it does not reveal the novelty of these results.

3. Recommend that the introduction (first three paragraphs) be rewritten to fit the target issue “Advanced packaging materials for food safety, storage and transportation.” I emphasize: “...packaging materials for...food products.”

4. Recommend discussing the food safety of this type of coating. There are no own studies, therefore, in the introduction, perhaps in materials and methods, it is imperative to find a place and reason in the discussion to provide information to prove the “food” safety of the materials used for packaging “food products”.

5. Section 3. Materials and methods. It is necessary to prove that nanocellulose was obtained using the described method; to do this, show compliance of the characteristics of the cellulose-containing product with the requirements of nanofibrillated cellulose (length, not diameter of fibers, degree of crystallinity, degree of polymerization) with reference not to one’s own publications, but to reviews introducing the term “nanofibrillated cellulose” . Your case describes a method for producing a fibrillated cellulose-containing product from hemp stems.

6. Recommend indicating the yield of fibrillated cellulose-containing hemp product in terms of hemp stems.

I looked at references 20 and 30, which refer to hemp stems as “waste.” Hemp stems are not hemp waste unless you consider hemp as a source of hemp oil.

7. Section 3. Materials and methods. It is recommended to indicate the quantitative content of fibrillated cellulose-containing product in each of the laminates.

8. Section Discussion of the results as presented does not allow us to make a decision on the required number of cellulose layers, since one layer does not provide continuity of the coating, and a larger number of layers leads to an opaque coating with unclaimed properties. This discussion needs to be rewritten.

8. In Figure 5, the water vapor transmission rate (WVTR) values decrease with increasing number of layers of fibrillated cellulose-containing product. Why does the description of the drawing contradict the picture presented? I quote lines 178-179: “In the case of laminates with a cellulose layer, a higher cellulose layer count increases the value of WVTR.”. Please note that in the annotation the opposite statement is for a laminate with a single layer of cellulose: “The introduction of a single NFC layer was shown to reduce WVTR by up to 5.5-fold.”.

9. Comparison of the results obtained with the results of source 16 indicates the lack of scientific novelty of the manuscript, since much more reasoned results were obtained in 2019 by adding commercial nanocrystalline cellulose to PBS with the effect of reducing the water vapor transmission rate (WVTR), pursuing the same goal of creation packaging for food products. I emphasize that integrating nanocrystalline cellulose inside a coating is much more difficult than making a layer-by-layer coating.

10. Unfortunately, the authors did not pay adequate attention to this source 16 in the introduction. This situation needs to be corrected.

11. Conclusions need to be redone. The manuscript does not justify the "single-layer coating" as the most effective, since there is a lack of continuity of the coating with fibrillated cellulose-containing product PBS, there is no information about the amount of fibrillated cellulose-containing product in the laminate, there is no information about the type of food product and the safety of the proposed laminate. On the same topic: in the manuscript there is no justification for the “high-barrier” properties of the proposed films, in the conclusion this word is not used at all.

Author Response

The experimental article "Biodegradable Poly(Butylene Succinate) Laminate with Nanocellulose Interphase Layer for High Barrier Packaging Films Application" describes the process of obtaining a multilayer coating intended for food products, and includes the results of ongoing research by these authors. The manuscript describes the preparation of a fibrillated cellulose-containing product from hemp stems, called “nanocellulose” by the authors, which is proposed to be applied layer-by-layer to PBS. As conceived by the authors and similar to the research of Xu [16. doi:10.1007/s10965-019-1783-8], the new laminate, while retaining all its valuable properties for food packaging, should acquire a new quality, specifically reduce the water vapor transmission rate and become “High Barrier Packaging Films”. Unfortunately, in the author’s interpretation, instead of commercial nanocrystalline cellulose and the technology of integration into PBS, it is proposed to introduce the resulting fibrillated cellulose-containing product from hemp stems with a main substance content of 89% into the coating using the “layer-by-layer” method. As it turned out, the effect of reducing the water vapor transmission rate can be achieved by applying from one to five layers, but at the same time the packaging loses its transparency and it is necessary to control the peeling of this packaging. One can recognize the honesty of the researchers-authors of the manuscript, but this manuscript in its presented form poorly corresponds to the Foods rating publication and requires serious revision. Below are comments and questions.

- Thank you for your comments and suggestions. We have implemented most of suggested changes, as well as worked hard to outline and better deliver key takeaways from our research. Notably, explaining why NFC was selected, why lamination is better choice than melt mixing, and measuring wt% of our NFC in the composite.

Comments and questions that need to be answered with the text of the article:

  1. It is recommended to include in the title of the article, in the annotation, and in conclusion, the definition of these films as “food”, specifically, “intended for food packaging.” Example of the title “Poly(butylene succinate) with an interfacial layer of nanocellulose as high-barrier food packaging films.”

- We have changed the abstract to reflect the intended application of food packaging. As research is still ongoing, we avoided claiming in the title that packing is suitable for food, as there are many tests that still need to be conducted. The title is already long, so we decided against adding extra words for “food packaging applications”.

  1. Recommend to indicate in the annotation the types of food products that can be packaged with the described coating. Proposal "NFC can serve as an effective barrier without the use of chemical modification methods." can be excluded because it does not reveal the novelty of these results.

- The abstract was enhanced, but we included suitable products in the introduction. We believe it is important to note that it was not chemically modified, as a lot of research involving NFC achieve excellent properties by employing some kind of modification or compatibilization.  

  1. Recommend that the introduction (first three paragraphs) be rewritten to fit the target issue “Advanced packaging materials for food safety, storage and transportation.” I emphasize: “...packaging materials for...food products.”

- The introduction section was enhanced, and some sections were rewritten or added.

  1. Recommend discussing the food safety of this type of coating. There are no own studies, therefore, in the introduction, perhaps in materials and methods, it is imperative to find a place and reason in the discussion to provide information to prove the “food” safety of the materials used for packaging “food products”.

- Introduction enhanced.

  1. Section 3. Materials and methods. It is necessary to prove that nanocellulose was obtained using the described method; to do this, show compliance of the characteristics of the cellulose-containing product with the requirements of nanofibrillated cellulose (length, not diameter of fibers, degree of crystallinity, degree of polymerization) with reference not to one’s own publications, but to reviews introducing the term “nanofibrillated cellulose” . Your case describes a method for producing a fibrillated cellulose-containing product from hemp stems.

- Depending on the source of the plant. Cellulose content can differ, just as treatment methods for different plant sources yield different sizes of fibrils. The procedures vary widely between the authors. We have referenced our previous paper that analyzed our used hemp and provided chemical composition results, crystallinity, and size analysis. For nanofibrillated cellulose, the diameter (width) of fibrils is commonly reported. Length for NFC is hard to determine due to the entangled structure of NFC, so it is rarely reported. To better characterize our produced fibrils, we have included an SEM image. We have added a reference to the fibrillation method.  

  1. Recommend indicating the yield of fibrillated cellulose-containing hemp product in terms of hemp stems.

I looked at references 20 and 30, which refer to hemp stems as “waste.” Hemp stems are not hemp waste unless you consider hemp as a source of hemp oil.

- Added to the section 2.2 sentence, “The NaOH treatment reduces the biomass weight by about 20 to 30%.”

- We have not presented these references in a that way. A matter about hemp stems and their waste status in this research article was not discuessed. But this is indeed a complex matter that is sometimes a bit confusing. The upper and middle parts of the stem are commonly processed to separate high-quality fiber bundles. The remaining part of these is mostly treated as waste, while the bottom part of the stem is commonly left on fields and not harvested at all. Hemp is treated as a multifunctional plant, but not all of its parts are equally utilized to meet the same demand. Thus, stems are still considered underutilized and still produce waste. We received this agricultural waste from local farmers.

  1. Section 3. Materials and methods. It is recommended to indicate the quantitative content of fibrillated cellulose-containing product in each of the laminates.

- Thank you for your suggestion. It is indeed very valuable information, which has been included in Materials section Table 1.

  1. Section Discussion of the results as presented does not allow us to make a decision on the required number of cellulose layers, since one layer does not provide continuity of the coating, and a larger number of layers leads to an opaque coating with unclaimed properties. This discussion needs to be rewritten.

- The discussion section was improved and rewritten for clarity. We added an optical image to the UV-Vis section to better show the impact of changes in transparency.

  1. In Figure 5, the water vapor transmission rate (WVTR) values decrease with increasing number of layers of fibrillated cellulose-containing product. Why does the description of the drawing contradict the picture presented? I quote lines 178-179: “In the case of laminates with a cellulose layer, a higher cellulose layer count increases the value of WVTR.”. Please note that in the annotation the opposite statement is for a laminate with a single layer of cellulose: “The introduction of a single NFC layer was shown to reduce WVTR by up to 5.5-fold.”.

- The section was rewritten for greater clarity. In addition, we show that all NFC composites showed decreased WVTR compared to laminates without NFC.

  1. Comparison of the results obtained with the results of source 16 indicates the lack of scientific novelty of the manuscript, since much more reasoned results were obtained in 2019 by adding commercial nanocrystalline cellulose to PBS with the effect of reducing the water vapor transmission rate (WVTR), pursuing the same goal of creation packaging for food products. I emphasize that integrating nanocrystalline cellulose inside a coating is much more difficult than making a layer-by-layer coating.

- The authors of this publication used melt mixing and reported improved barrier properties for composites with 3 wt% of NCC. The authors did not use blown films but only compression-molded films, which are not suitable methods for large-scale packing production. In accordance with your suggestion No. 7, We have included the wt% of NFC in our films, showing the best results with only 0.35 wt% of NFC. NCC is about 3–5 times more expensive than NFC in industrial production. We do agree that melt mixing is generally easier to integrate, but it also requires much more nanocellulose. 

  1. Unfortunately, the authors did not pay adequate attention to this source 16 in the introduction. This situation needs to be corrected.

- We chose not to discuss this work in detail in our introduction section. The authors used NCC and chitin nanowhiskers, while we used NFC. NCC is about 3–5 times more expensive than NFC in industrial production. The authors reported barrier properties only for composites with 3 wt% NCC and for composites with additional methylene diphenyl diisocyanate. The authors used melt mixing instead of coating. In accordance with your suggestion No. 7, We have included the wt% of NFC in our films, showing the best results with only 0.35 wt% of NFC. We have enhanced the introduction section to better reflect our choice of NFC. In addition, this reference is also included in the discussion of WVTR data.

  1. Conclusions need to be redone. The manuscript does not justify the "single-layer coating" as the most effective, since there is a lack of continuity of the coating with fibrillated cellulose-containing product PBS, there is no information about the amount of fibrillated cellulose-containing product in the laminate, there is no information about the type of food product and the safety of the proposed laminate. On the same topic: in the manuscript there is no justification for the “high-barrier” properties of the proposed films, in the conclusion this word is not used at all.

- Thank you for your suggestions. Conclusions have been improved.

Reviewer 3 Report

Comments and Suggestions for Authors

The article by Nabels-Sneiders M. et al. addresses the preparation of barrier films made of poly(butylene succinate) containing a nanocellulose intermediate layer for potential application as biodegradable packaging for the food industry. The authors obtain films containing different thicknesses of the intermediate nanocellulose layer, study the morphology of this layer by optical, electron, and atomic force microscopy, and then measure the adhesive strength of the resulting laminate, its transmittance in the UV-visible range, weight loss upon heating, and water vapor transmission rate. The article is perfectly prepared.

Specific comments are as follows.

Line 217: “It has a melting point”. It would be valuable if the authors provided the molecular weight and dispersity of poly(butylene succinate).

Line 234: “the average diameter of the nanofibrils is 70 nm”. The authors produced microfibrillated cellulose rather than nanofibrillated one (see 10.1007/s10570-014-0357-5, 10.1016/j.carbpol.2023.120896), i.e., it is MFC rather than NFC whose diameter is significantly smaller (4–10 nm).

Line 246: “The concentration of NFC suspension was adjusted to 0.8 w%”. The continuous medium of the suspension must be specified (water or some more volatile organic solvent).

Line 262: “the samples were mounted on electrically conductive double-sided carbon tape”. It should be indicated whether the authors used sputtering of any metal (gold, silver, etc).

Line 277: “A SolidSpec3700 UV-VIS-NIR Shimadzu (Kyoto, Japan) spectrophotometer”. Was the thickness of films containing different nanocellulose contents different? If the films were of different thicknesses, did the authors take this into account when obtaining the spectra?

Author Response

The article by Nabels-Sneiders M. et al. addresses the preparation of barrier films made of poly(butylene succinate) containing a nanocellulose intermediate layer for potential application as biodegradable packaging for the food industry. The authors obtain films containing different thicknesses of the intermediate nanocellulose layer, study the morphology of this layer by optical, electron, and atomic force microscopy, and then measure the adhesive strength of the resulting laminate, its transmittance in the UV-visible range, weight loss upon heating, and water vapor transmission rate. The article is perfectly prepared.

- Thank you for the suggestions. We have improved our manuscript.

Specific comments are as follows.

Line 217: “It has a melting point”. It would be valuable if the authors provided the molecular weight and dispersity of poly(butylene succinate).

- The used PBS is commercial-grade and not synthesized by our team. Unfortunately, the manufacturer does not provide this data. In accordance with the literature, PBS with such high properties should have a molecular weight above 100 000 and close to 200 000 [https://doi.org/10.1002/biot.201000136]. This information would be helpful, but at the moment we do not have equipment to measure these values. As it is widely available, researchers can access it and measure it on their own.

Line 234: “the average diameter of the nanofibrils is 70 nm”. The authors produced microfibrillated cellulose rather than nanofibrillated one (see 10.1007/s10570-014-0357-5, 10.1016/j.carbpol.2023.120896), i.e., it is MFC rather than NFC whose diameter is significantly smaller (4–10 nm).

- There have been various definitions for the nanofibrillated cellulose diameter range. The widely cited journal Chemical Reviews has an article defining it with diameters up to 100 nm (https://doi.org/10.1021/acs.chemrev.7b00627), while one more widely cited paper from carbohydrate polymers (https://doi.org/10.1016/j.carbpol.2013.08.069) defines it under 60 nm, and yes, there are more papers that place it in the range of up to 100 nm (https://doi.org/10.3389), while indeed some are defining the diameter maximum at 20–30 nm.

- We have provided an SEM image of our NFC. And we have noted in Section 2.2. that our NFC can also be considered micro/nanofibrillated cellulose (MNFC).

Line 246: “The concentration of NFC suspension was adjusted to 0.8 w%”. The continuous medium of the suspension must be specified (water or some more volatile organic solvent).

- Changed to “The concentration of NFC/water suspension was adjusted to 0.8 w% after process optimization”

Line 262: “the samples were mounted on electrically conductive double-sided carbon tape”. It should be indicated whether the authors used sputtering of any metal (gold, silver, etc).

- Added line “No sputter coating was applied.”

Line 277: “A SolidSpec3700 UV-VIS-NIR Shimadzu (Kyoto, Japan) spectrophotometer”. Was the thickness of films containing different nanocellulose contents different? If the films were of different thicknesses, did the authors take this into account when obtaining the spectra?

- Device does not adjust results to sample thickness. We selected samples with the closest thickness. NFC makes samples slightly thicker. We have enhanced the discussion to reflect this.

Reviewer 4 Report

Comments and Suggestions for Authors

This research explored poly (butylene succinate) (PBS) as the polymer matrix, coated with nano-fibrillated cellulose (NFC) with different layers, and successfully prepared the packaging film material with high barrier property. Water vapor transmittance (WVTR) was tested for two months in a controlled environment at 38 ° C and 90% relative humidity. The introduction of a single NFC layer can reduce WVTR values by up to 5.5-fold. The number of high floors with more than 10 layers reduces the adhesion of the laminate. Morphological studies show that with the increase of the number of NFC layers, there is a significant difference. Experiments have shown that NFC can be used as an effective barrier without the use of chemical modification methods. In terms of sustainable and renewable methods, the laminate produced is a promising alternative to traditional packaging.

In my view, there are some questions need to be solved. Here is the detail of necessary revision,

1.     It is suggested to add references to the analysis of the UV transmittance characterization experiment.

2.     In the introduction part of the article, it is suggested to add the specific application of PBS-NFC coated composite materials.

3.     The biodegradability of the material is one of its highlights, and it is suggested to add an experiment on the biodegradability.

4.     In the mechanical characterization experiment, the adhesion of the composite was evaluated. Did the author consider exploring other mechanical properties such as its extensibility?

Author Response

This research explored poly (butylene succinate) (PBS) as the polymer matrix, coated with nano-fibrillated cellulose (NFC) with different layers, and successfully prepared the packaging film material with high barrier property. Water vapor transmittance (WVTR) was tested for two months in a controlled environment at 38 ° C and 90% relative humidity. The introduction of a single NFC layer can reduce WVTR values by up to 5.5-fold. The number of high floors with more than 10 layers reduces the adhesion of the laminate. Morphological studies show that with the increase of the number of NFC layers, there is a significant difference. Experiments have shown that NFC can be used as an effective barrier without the use of chemical modification methods. In terms of sustainable and renewable methods, the laminate produced is a promising alternative to traditional packaging.

- Thank you for your comments and suggestions.

In my view, there are some questions need to be solved. Here is the detail of necessary revision,

  1. It is suggested to add references to the analysis of the UV transmittance characterization experiment.

- The discussion was enhanced, and two references were added.

  1. In the introduction part of the article, it is suggested to add the specific application of PBS-NFC coated composite materials.

- The introduction was improved.

  1. The biodegradability of the material is one of its highlights, and it is suggested to add an experiment on the biodegradability.

- We have extensively tested the biodegradation of PBS and PBS/cellulose composites in our previous studies and have found that PBS films degrade in about 70–80 days in composting conditions. As the samples are pure PBS films from the outside and the NFC layer is relatively small, we expected that it would degrade in a similar manner as the pure PBS structure. As PBS films have not been altered and biodegradation is mainly a surface process, there should be no significant difference.

  1. In the mechanical characterization experiment, the adhesion of the composite was evaluated. Did the author consider exploring other mechanical properties such as its extensibility?

- We tested tensile properties, and the results did not reveal notable changes from neat PBS films. Thus, we chose not to include this data. We plan to make further studies on the durability of these materials, including improving adhesion. 

Round 2

Reviewer 2 Report

Comments and Suggestions for Authors

The experimental article “Biodegradable Poly(Butylene Succinate) Laminate with Nanocellulose Interphase Layer for High Barrier Packaging Films Application”, after the reviewer’s comments, retained its name, but underwent a number of serious changes. In particular, more than half of the comments were corrected by the authors. The authors answered a number of questions to the reviewer with detailed arguments.

Author Response

There seem to be no comments to answer.

Reviewer 4 Report

Comments and Suggestions for Authors

 Accept in present form

Author Response

There seem to be no comments to answer.